# Human Attitude toward Reptiles: A Relationship between Fear, Disgust, and Aesthetic Preferences

**DOI:** 10.3390/ani9050238

**Published:** 2019-05-14

**Authors:** Markéta Janovcová, Silvie Rádlová, Jakub Polák, Kristýna Sedláčková, Šárka Peléšková, Barbora Žampachová, Daniel Frynta, Eva Landová

**Affiliations:** 1Department of Zoology, Faculty of Science, Charles University, Viničná 7, 128 43 Prague, Czech Republic; 2National Institute of Mental Health, Topolová 748, 250 67 Klecany, Czech Republic

**Keywords:** reptiles, emotions, fear, disgust, beauty

## Abstract

**Simple Summary:**

Although there are many articles about reptiles, no one has ever studied the human perception of reptiles as a whole, a group that would include representatives of different taxonomic clades. Thus, we designed a study of human perception of all reptiles focusing on the relationship between perceived fear, disgust, and aesthetic preferences. Respondents evaluated various reptile images and the results revealed that people tend to perceive them as two clearly distinct groups based on their similar morphotype—legless reptiles (incl. snakes) and other reptiles with legs. In the case of snakes, the most feared species also tend to be perceived as beautiful. Compared to the most feared reptiles with legs (lizards, turtle, crocodiles), the legless once tend to be perceived as more disgusting. In both groups, species perceived as the least beautiful were the same as those rated as the most disgusting. Thus, reptiles cannot be rated as both beautiful and disgusting at the same time.

**Abstract:**

Focusing on one group of animals can bring interesting results regarding our attitudes toward them and show the key features that our evaluation of such animals is based on. Thus, we designed a study of human perception of all reptiles focusing on the relationship between perceived fear, disgust, and aesthetic preferences and differences between snakes and other reptiles. Two sets containing 127 standardized photos of reptiles were developed, with one species per each subfamily. Respondents were asked to rate the animals according to fear, disgust, and beauty on a seven-point Likert scale. Evaluation of reptile species shows that people tend to perceive them as two clearly distinct groups based on their similar morphotype. In a subset of lizards, there was a positive correlation between fear and disgust, while disgust and fear were both negatively correlated with beauty. Surprisingly, a positive correlation between fear and beauty of snakes was revealed, i.e., the most feared species also tend to be perceived as beautiful. Snakes represent a distinct group of animals that is also reflected in the theory of attentional prioritization of snakes as an evolutionary relevant threat.

## 1. Introduction

### 1.1. Focused on reptiles

Reptilia represents an ancient and diversified group of vertebrates that includes Testudines (turtles, tortoises and terrapins) and Diapsida, which further splits into Lepidosauria (comprising Squamata and Rhynchocephalia) and Archosauria (comprising Crocodylia and Aves). The precise position of Testudines within Diapsida is uncertain [1]. They may be a sister group of either lepidosaurs [2] or archosaurs [3]. Reptilia is diversified group of animals with unclear phylogeny of major clades, comprising turtles, crocodiles, tuatara, lizards, snakes and birds.

The birds, however, differ so much from the other groups that they are almost always recognized and categorized as a separate category by humans, both in traditional [4] and ethnobiological classification [5]. Because this paper is focused on human perception of these animals, we will use the term “reptiles” in the same manner, i.e., as a paraphyletic group of Reptilia excluding birds and extinct species.

Approximately 10,885 reptile species are currently described, and this number grows each year as others are added. For example, in 2018, 157 new species were described—113 lizards, 39 snakes, 4 worm lizards and one turtle (data valid to 14th November 2018; [6]). Nevertheless, in some aspects, reptiles still represent a neglected group of vertebrates, especially when compared to birds and mammals [7,8]. Because of that, basic information is lacking for a substantial number of species (and not only the newly discovered ones), be it the population size, life-history, or real distribution. These gaps in knowledge also present a complication in the protection of endangered species, because it is not possible to determine whether and by what means a particular species is threatened [9,10].

### 1.2. Research of Human Relationship to Animals

A similar bias as in our knowledge about reptile species can be found when researching people’s relationship to animals. It can be explored from many different perspectives, e.g., in terms of folk traditions, utilitarian reasons, negative attitude, pet keeping or nature protection [11,12,13,14]. The whole variety of human interactions with local fauna has been traditionally the subject of ethnozoology (or specifically ethnoherpetology when focused on reptiles), a discipline that has thrived since the 19th century [15]. Recently, there is a growing body of evidence that ethnozoology through studying people’s perception of animals is indispensable in sustainable management of natural resources and might, therefore, play a crucial role in conservation of endangered species [16], including reptiles [17,18,19], protection of which is complicated by the aversion they often elicit in people [20].

Reptile studies are often unbalanced in terms of the choice of stimuli that respondents rate, either as images or words, and the majority of them focus just on snakes in regards to snake fear and phobias (see *Negative relationship to animals*). This can be emphasized especially when compared to studies of human attitude toward other animals (both vertebrates and invertebrates [21,22]). For example, several mammalian species from different families have been selected to study human relationship, e.g., the lion (*Panthera leo*), chimpanzee (*Pan troglodytes*), elk (*Alces alces*) or dolphin (*Tursiops truncatus*) [21,23], although there is nearly half as many mammals as reptiles (5,792 species of mammals are currently recognized [24]). Conversely, a group of reptiles is most often represented by only a single taxonomic category, i.e., the lizards (including snakes), most often further unspecified [22] or represented by the common sand lizard (*Lacerta agilis*) and grass snake (*Natrix natrix*) or some pythons (*Python* sp.) [12,21,23]. Only a few studies also include crocodiles or turtles [22,25]. Thus, the selection of reptilian species in the human-animal relationship research does not adequately reflect their morphological diversity. The only exception can be found in studies investigating the relationship of people to the local fauna, which usually includes the most important reptilian species in the area [13,14,26].

To our knowledge, no one has ever studied the human perception of reptiles in whole, as a category that would include representatives of different taxonomic clades (i.e., turtles and tortoises, lizards, crocodiles, and tuatara) reflecting their large-scale biological diversity. Yet focusing on one particular group of animals can bring interesting results regarding our attitudes toward them and show by what rules evaluation of such animals is conducted. For example, mammals are evaluated mainly by body shape and coat pattern. Animals with a distinctive pattern or longer and denser hair that resemble plush toys are positively perceived. Conversely, people perceive negatively subterranean species with stunted eyes or faintly colored hair. These properties affect the evaluation of mammals across taxonomic groups [27,28]. Similarly, people evaluate birds based on their body shape (head size in relation to the body, tail length, limb length, [29]), and pattern [30]. Colors have minor effect when evaluating birds across all main taxa (families), but affect preferences within some groups, such as parrots [31] or pittas [30].

### 1.3. Negative Relationship to Animals

Ekman and Cordaro [32] distinguish at least seven basic emotions that are shared between different cultures everywhere, namely anger, fear, surprise, sadness, disgust, contempt and happiness. These basic emotions are described as discrete, automatic answers to important life situations that have helped human ancestors to survive in the past. From an evolutionary perspective, fear and disgust, in particular, represent a biologically adaptive way of responding to situations that may be potentially life-threatening. Fear is triggered in the presence of a predator or other significant fear stimulus, under the influence of the sympathetic nervous system, initiating a specific and rapid defensive behavioral reaction known as "fight or flight [33]. Disgust then acts as a mechanism to protect the body from potential contamination and disease and motivates disease-avoidance behavior [34,35]. Rozin and Fallon [36] and Rozin et al. [37] assume that disgust originated as a mechanism to reject poorly tasting food that could be poisonous, spoiled, or otherwise degraded. Fear and disgust also play a role in anxiety disorders, including specific animal phobias [33,38,39,40].

Of all reptiles, only certain groups, especially snakes and crocodiles, are really dangerous for humans. Crocodiles may be deadly due to their impressive body size and predatory way of life [41,42,43]. Certain snakes, on the other hand, can either produce potent venom capable of killing an adult human (Viperidae, Elapidae; [44]) or become dangerous when reaching critical size, especially large constrictors like the reticulated python (*Malayopython reticulatus*, [45]) or green anaconda (*Eunectes murinus*, [46] (p.131–164)). According to some authors e.g., [47,48], humans and snakes have a long predator-prey co-evolutionary history and fatal attacks of venomous and constrictor snakes on our direct ancestors during their early development in Eastern Africa have shaped the relationship to this widely feared group of animals [49]. A significant danger was mainly represented by highly venomous snakes of the viper family (Viperidae, [6]) that lived in the same area of human origins [50,51]. Thus, it was important for early humans to recognize and react appropriately to this threat. There is an extensive line of evidence suggesting that even nowadays, when the risk of snake encounters has much reduced, snakes still represent a significant stimulus for humans attracting increased attention. This phenomenon is supported by studies showing a rapid detection of snakes compared to inanimate objects, such as flowers and mushrooms [52,53,54] or other animals [55,56,57], mediated by specific neural mechanisms [58,59,60,61,62]. This was reported in both humans and non-human primates [63,64]. For these reasons, it is rational to believe that snakes, although taxonomically a part of lizards, represent a different cognitive category for humans. As a substantial part of this study, we aimed to examine this specific position of snakes in relation to fear, but also to disgust and beauty (as a positive dimension to contrast with fear).

As for disgust, reptiles do not represent a significant group. Studies have shown that people are particularly repelled by invertebrates or some rodents (mice or rats) [65,66,67,68]. Conversely, for humans, reptiles may represent a significant source of food rich in proteins, especially in warmer areas [13,69,70]. Disgust (together with fear) has only been studied in relation to snakes where it may play a significant role in the development of clinical fear in phobics [71,72].

### 1.4. Positive Relationship to Animals

Animals can also generate positive emotions in humans and be interesting for them from an aesthetic point of view [73]. In literature, terms, such as "beauty", "aesthetic value" or "attractiveness" are often confused or considered as synonymous [74,75,76]. Still, it is important to correctly define what we are asking about when evaluating animals by humans. For example, the attractiveness of an animal may not always be related to positive assessment. An animal that seems attractive to humans can be somewhat bizarre, atypical or obscure [77]. Thus, with a positive attitude towards animals, the concepts, such as likeness, pleasantness and beauty are more likely to be combined, so the animals evaluated in this way inspire positive experiences [78]. The beauty of animals is an aesthetic value that is perceived consistently between different cultures and ethnicities [30,79] and both sexes at the same time [80].

Aesthetic preferences for animals (i.e., the beauty of animals perceived by humans) were initially examined particularly in relation to the visits of zoos. Results of these studies were to help zoos adapt their concept to the taste of visitors [81,82,83,84]. Other research focused on beauty in relation to the conservation of endangered species [85,86], people were more willing to contribute to helping animals if they considered them as more beautiful [87]. In the context of people's willingness to protect beautiful species, further work has focused on the analysis of animals in human care (zoological gardens) and their aesthetic evaluation. In mammals, their beauty has been shown to influence the number of individuals kept [27], the same results were found for parrots [81], turtles [88] and boas (now Boidae and Pythonidae; [89]). Understanding the cues behind people's positive attitudes in a particular group of animals can help in planning conservation programs for endangered species. Beautiful animal species can serve as flag species to support the conservation of a given habitat [90,91]. Conversely, in the case of animals that are not rated as beautiful, it is advisable to use other ways to increase the interest in those species when planning a campaign. 

### 1.5. Aims

Based on the published studies on reptiles and the relationship of people to this group, we asked the following questions:What is the human attitude toward a group of reptiles in terms of positive and negative emotions and does our evaluation rely on basic morphotypes, specifically, are legless snakes perceived differently than the rest of reptiles?What is the relationship between perceived fear, disgust, and beauty of reptiles? Do these evaluations affect each other?

## 2. Materials and Methods

### 2.1. Selection and Preparation of Stimuli

To test the relationship of people to reptiles, we have prepared two sets of pictures that cover a wide variability of the reptile morphology: At least one species representing the main lineages was used. These sets were designed to cover the wide variability while retaining a reasonable number of stimuli that could be rated by respondents without substantial tiredness. We selected one representative from each existing reptile subfamily in each set, i.e., 127 stimuli per set (see [92]; the most comprehensive study of reptile taxonomy at the time of experiment planning). For a full list of included species, see Appendix A. The second set containing different species (except monotypic subfamilies) was created to test whether selecting different subfamily representatives will lead to the same results. For each selected species, we found a representative photo of an adult individual on the Internet or among our own resources. Only photos in suitable resolution (at least 800 × 533 pixels) depicting the animal in full body were chosen. We adjusted the photographs to a standardized form, i.e., the animals were placed on a white background, adjusted to a similar position and comparable body size.

### 2.2. Testing Emotional Response to Reptiles

We uploaded both sets of standardized photos to a special web application available at www.krasazvirat.cz (Figure 1). We had the sets evaluated by Czech respondents (aged 18–88) separately for perceived fear, disgust and beauty on a seven-point Likert scale (7 corresponded to the strongest fear/disgust or the most beautiful species, 1 corresponded to the lowest response; [93]). Before the evaluation, each respondent filled in a short questionnaire (concerning the age, gender, and type of education), was informed about the content of the experiment and provided his/her consent to the processing of personal data (all in the Czech language). Some respondents evaluated the sets by all of the measured dimensions, however, because there was a reasonable time delay between each of the evaluations (several months), we considered them independent. Each of the sets (Set 1 and 2) were evaluated by different respondents: Set 1 evaluated 122 respondents by perceived fear, 126 by disgust, and 157 by beauty; Set 2 evaluated 184 (fear), 143 (disgust), and 188 (beauty) respondents.

### 2.3. Statistical Analyses

In order to quantify and test the congruence in species ranking provided by different respondents, we adopted a two-way, consistency, average-measures intra-class correlation (ICC; [94,95] computed in R (irr package). Principal component analysis (PCA) was performed to visualize the multivariate structure of the data sets and to extract uncorrelated axes for further analyses. MANOVA was applied to test the effects of independent explanatory variables. Mann-Whitney test was used as a non-parametric alternative for variables deviating from normality (raw scores). Most of the calculations were performed in R [96] and Statistica 9.1. [97].

### 2.4. Ethical Note

This study was carried out in accordance with the recommendations of Institutional Review Board (IRB), Faculty of Sciences, Charles University approval n. 2013/7. All subjects gave their written informed consent in accordance with the Declaration of Helsinki.

## 3. Results

### 3.1. Agreement among Respondents

Results of the ranking procedure revealed considerable congruence among the respondents. Both the reliability of the individual rankings (ICC = 0.562, 0.441, 0.417 for fear, disgust, and beauty rankings, respectively, with all *p* < 0.001) and the ICC for the average-measures were in an excellent range (ICC = 0.994, 0.99, and 0.991 for fear, disgust, and beauty rankings, respectively; [98]. These results indicate that there was a high degree of agreement within the groups of respondents and suggest that the emotions elicited by reptiles and their beauty were rated similarly. The results for Set 2 were comparable; single ratings: ICC = 0.579, 0.416, 0.364 for fear, disgust, and beauty rankings, respectively, and average ratings: ICC = 0.996, 0.99, 0.991 for fear, disgust, and beauty rankings, respectively.

The multivariate analysis of variance (MANOVA) revealed no effect of gender, age, profession, nor their interaction on the rankings of fear and disgust of Set 1. In beauty rankings, small effects of gender (Wilks = 0.0629, F_45,127_ = 2.23, *p* = 0.0232) and age (Wilks = 0.0485, F_45,127_ = 2.94, *p* = 0.0045) were found. To identify the species that substantially contributed to these differences, we performed Mann–Whitney U tests comparing the raw ranks of each species in male/female and younger (18–30, *n* = 85)/older (31–73, *n* = 68) respondents (Bonferroni-corrected). However, these tests revealed no differences in beauty rankings between both genders, and only one difference in the Chinese butterfly lizard (*Leiolepis reevesii*) that was rated as more beautiful by older respondents.

Similar results were found in Set 2: MANOVA revealed a small effect of gender*age interaction (Wilks = 0.2021, F_50,127_ = 1.55, *p* = 0.0387) on the rankings of fear and a small effect of age (Wilks = 0.0173, F_9,127_ = 4.02, *p* = 0.0137) on the rankings of disgust, but Bonferroni-corrected Mann-Whitney U tests revealed no differences in rankings between the two age groups (18–30, *n* = 106; 31–88, *n* = 37) in any of the species. In beauty rankings, small effects of gender (Wilks = 0.2180, F_54,127_ = 1.53, p = 0.0405) and age (Wilks = 0.1220, F_54,127_ = 3.06, *p* < 0.0001) were found. Similarly to Set 1, only two differences between the two age groups (18–30, *n* = 111; 31–73, *n* = 77) were found: Older people ranked the Central American river turtle (*Dermatemys mawii*) and yellow-spotted Amazon turtle (*Podocnemis unifilis*) as more beautiful. Because the differences among respondents of various groups were very small, we decided to pool the data for each ranking of each set in further analyses concerning the means or multivariate axes (PCA) computed from the preference ranks. Both of these methods extract the agreement among respondents and thus further blend the minor effects of gender and/or age.

### 3.2. Multivariate Analyses of Datasets

In Set 1, PCA based on the fear rankings generated 121 unconstrained axes, 27 of which were of an eigenvalue higher than 1. However, most of the variability was explained by the first axis (69.68%), which was also highly correlated with the mean rankings of fear (Spearman r^2^ = 99.5%). Compared to that, the explanatory power of the other axes was much lower (9.98%, 4.41%, 3.27% for PC2, PC3 and PC4, respectively), gradually decreasing down to <0.001% in PC84 to PC121. PCA analyses based on the disgust and beauty rankings, as well as the data from Set 2 showed a very similar pattern (see Table 1). Because of that, we only extracted the scores of the first two axes of each ranking/Set and further analyzed their mutual correlations (Spearman, see Table 2).

As the next step, we performed a principal component (PC) analysis of the pooled data set of fear, disgust and beauty rankings, separately for Set 1 and 2 (for the numbers of axes, eigenvalues and explanatory power, see Table 1). Plotting the species scores onto PC1 and PC2 axes yielded very interesting results (see Figure 2): The reptiles are divided into two clearly separate groups, one consisting of snakes, legless lizards, and worm lizards (“legless reptiles”), and the other of lizards, turtles, and a tuatara (“other reptiles”). Notable genera are *Larutia* and *Lygosoma*, which possess visible legs with a long, snake-like body, and thus are seen and ranked by human respondents as something “in between” the legless and other reptiles. These results show that snakes, together with other legless reptiles, are indeed special and are ranked differently by humans: They evoke higher fear and disgust than other reptiles.

Similarly, when we plotted the species scores onto PC1 and PC2 axes of the PC analyses done separately for fear, disgust, and beauty rankings (see Figure 3), the pattern followed the segregation into the two main groups as described above, with a few exceptions specific for the particular ranking types. The most pronounced difference was that in the case of fear, crocodiles formed a distinct group. Moreover, the Komodo dragon (*Varanus komodoensis*, Set 2) split from the main group of “other reptiles” and joined closer to the crocodiles. The alligator snapping turtle (*Macrochelys temminckii*, Set 1) split off from the other Testudines and joined closer to the crocodiles, but still remained within the main group of “other reptiles”. In the case of disgust and beauty, the crocodiles became a part of the “other reptiles” group, suggesting that they mainly evoke fear, but are not exceptional animals when considering disgust and/or beauty.

### 3.3. Mutual Correlation of the Fear, Disgust, and Beauty Ranks

To see the mutual relationship between the rankings, we plotted the correlations of the mean ranks. To examine whether snakes follow a different pattern, we did these analyses separately for snakes (Serpentes) and other reptiles (see Figure 4). We found that in both cases, disgust and beauty rankings closely negatively correlated (snakes—70.32% and 80.18% in Set 1 and 2, respectively; other reptiles—80.24% and 80.06%). Interestingly, correlations involving fear (fear x disgust, fear x beauty) in snakes differed from those in other reptiles: Their positivity/negativity was inverted. The correlation of fear and beauty was negative in other reptiles (34.06% and 31.26%), but positive in case of snakes (34.94% and 39.64%). Correspondingly, the correlation of fear and disgust was positive in other reptiles (57.34% and 50.30%) and negative in snakes (21.41% and 28.51).

## 4. Discussion

### 4.1. Relationship to Reptiles as a Group

Our results confirmed that people tend to separate snakes and legless lizards from other Sauria and the rest of the reptiles, i.e., the turtles, tuatara, and the crocodiles. This pattern was present on all examined dimensions; i.e., the respondents separated snakes from other reptiles when evaluating fear, disgust, and beauty. Interestingly, lizards with a snake-like body shape and visible, though very short limbs (*Larutia* and *Lygosoma*), form a transition between the two groups (see Figure 2). Thus, we may hypothesize that people discriminate reptiles into snakes and legless lizards on one side and other species on the other solely based on their external resemblance. As previously shown in a study by Alves et al. [26], when evaluating a set of pictures of local snakes and three worm lizards, the subjects called the whole set as “snakes”, disregarding the worm lizards as another taxonomic group. Thus, in human eyes, snakes represent a distinct group of animals, which is also reflected in the theory of attentional prioritization of snakes as an evolutionary relevant threat [47,48,99]. Moreover, people tend to extend this group even to other animals that are not snakes per see, but at least look like them. This is not so much surprising as in terms of survival, it is always better to respond with fear to an otherwise harmless animal that resembles a snake, than to underestimate a potentially serious threat.

### 4.2. Evaluation of Perceived Fear, Disgust, and Beauty

When reptiles are evaluated on specific dimensions (fear, disgust, and beauty), the two previously mentioned groups are almost always formed. The only exception is the evaluation of fear, in which crocodiles form a separate group. Morphologically, they fall within the category of other lizards with legs, but compared with them, crocodiles trigger more intense fear in humans. Such a result fits well with the fact that crocodiles together with snakes are the only reptilian predators capable of killing a human [41,42,43,100]. There is also the alligator snapping turtle (*Macrochelys*
*temminckii*) and Komodo dragon (*Varanus komodoensis*), which may be potentially dangerous for humans considering their large body size, although attacks from these species are very rare and the inflicted injuries are mostly manageable [101]. In fact, contrary to crocodiles or large constrictors, both the snapping turtle and Komodo dragon do not consider humans as prey. It is thus possible that these animals evoke high fear in the respondents not due to their real dangerousness, but rather due to their physical appearance, which may include large body size, sharp edges, and dark color.

The shape of an animal allometrically changes with changes in the body size. Because of this, human respondents are able to estimate size [27,28,29,30,31,79,80] and age (on the level of young/adult; [102]) of animals even when presented on pictures standardized for size. A similar effect was observed in this study. Larger reptile species were evaluated as more frightening, including the Komodo dragon. Moreover, as shown on snake studies, the posture of the animal can affect both rapid detection [103] and evaluation of fear [30]. In this study, the alligator snapping turtle was pictured with an open mouth, showing its sharp beak, which might have elicited a feeling of a threat within the respondents. Another explanation might be that the turtle was rated as frightening because of a general perceptual bias towards sharp objects; it has been reported in the literature that people rate sharp objects as more dangerous [104] and also activate a greater neural response in the amygdala when watching sharp objects [105].

Within the group of reptiles with the snake-like body form, we can notice that even though the worm and legless lizards morphologically belong to snakes, they are evaluated as less frightening, similarly to harmless snakes, e.g., the Madagascar blind lizard (*Xenotyphlops mocquardi,* nowadays synonymized with *Xenotyphlops grandidieri*, [106]).

Disgust ratings do not lead to a separation of another special category. In this case, crocodiles, snapping turtles, and Komodo dragons are grouped together with the other reptiles. Again, we can identify two species, the three-banded Larut skink (*Larutia trifasciata*) and banded supple skink (*Lygosoma haroldyoungi*) that, given their morphotype, may fall between the two groups. All worm lizards and legless lizards together with snakes possessing a worm-like body form were evaluated as the most disgusting ones within the snake group. These animals may remind people of some invertebrates (earthworms, larvae, parasitic worms) that usually elicit great disgust [68,107].

Based on perceived beauty, the discrimination of reptiles into two separate groups is similar to that of disgust. Surprisingly, the least beautiful species within the group of other reptiles was a turtle. Turtles are generally considered as popular animals, and this is also reflected in the number of turtle species kept in the zoos [88]. In our study, it was two representatives of the same subfamily Trionychinae, specifically the Malayan soft-shelled turtle (*Dogania subplana*) and Peacock soft-shelled turtle (*Nilssonia hurum*), which is remarkable by its atypical soft-shell carapace that deviates from usual turtle morphotype. Within the snake group, species perceived as the least beautiful are the same as those rated as the most disgusting. This points to the conclusion that reptiles cannot be rated as both beautiful and disgusting at the same time (see below).

### 4.3. Mutual Relationship of the Ranks

Intuitively, one might assume a negative correlation between negative emotions (such as fear and disgust) and beauty, which is closely related to the positive emotion of joy [108,109]. Our results show that this is true in the case of the beauty x disgust correlation: In both snakes and other reptiles, there is a close negative correlation of disgust and beauty. These results suggest that both ranks form two opposite sides of one axis; a lizard or snake that is ranked as beautiful cannot simultaneously evoke disgust in respondents [30,88,110]. Similarly, a negative correlation can be found between fear and beauty, but only in the case of non-snake reptiles. In snakes, this relationship is inverted, and snakes that evoke high fear are simultaneously ranked as partially beautiful/less disgusting. One reason behind this may be that dangerous snakes often possess saturated or aposematic coloration, which is perceived as beautiful [30,88,111]. Another explanation is that the snakes ranked as non-dangerous, i.e., evoking low levels of fear, may resemble earthworms and/or parasitic worms in appearance, thus evoking disgust [68,111]. Either way, snakes represent an exceptional group in terms of human perception.

## 5. Conclusions

Firstly, snakes are such important and specific animals that they are completely separated from the other reptiles as regards our perception of them, especially fear. We hypothesize that it is their peculiar legless body plan that guides the human categorization process. Moreover, the same pattern expands onto other species resembling snakes, the worm lizards (Amphisbaenia) and legless lizards.

Secondly, disgust and beauty together form one negatively correlated axis which is common to all reptiles. It is practically impossible for an animal to be both beautiful and disgusting at once. For fear, it is different. Fear correlates with other scales only partly and is independent on disgust and beauty forming a separate axis. Moreover, the polarity of the correlation of fear with beauty is inverted for snakes and non-snakes. Thus, a snake can be beautiful and dangerous at the same time, e.g., the Jerdon’s pitviper (*Protobothrops jerdonii*). Conversely, it is possible to find a non-snake reptile that is both disgusting and fearful, such as Borneo earless monitor *Lanthanotus borneensis*.

Thirdly, our results might be used in conservation biology efforts invested in the management of reptile populations threatened by extinction. It has been previously shown that especially in the case of reptiles, human perception and attitude plays a significant role in the effectivity of local conservation programs. Reptiles, as opposed to other vertebrates, such as mammals or birds, are not very popular and aversion they trigger might be a barrier to their protection. Here we show that a particular species with the snake-like body form are associated with intense fear and disgust and therefore, this group might be susceptible to more human attacks and lesser conservation efforts. Previous research has shown that these negative emotions reflected in aversive attitudes might be overcome through various educational programs targeted on people living in the specific locality.

## Figures and Tables

**Figure 1 animals-09-00238-f001:**
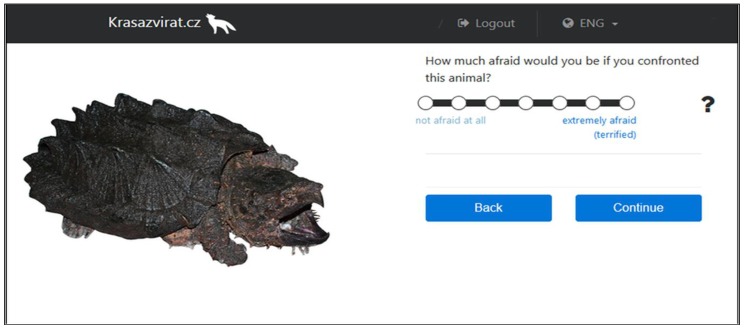
Sample of graphic design of web application for evaluating pictures.

**Figure 2 animals-09-00238-f002:**
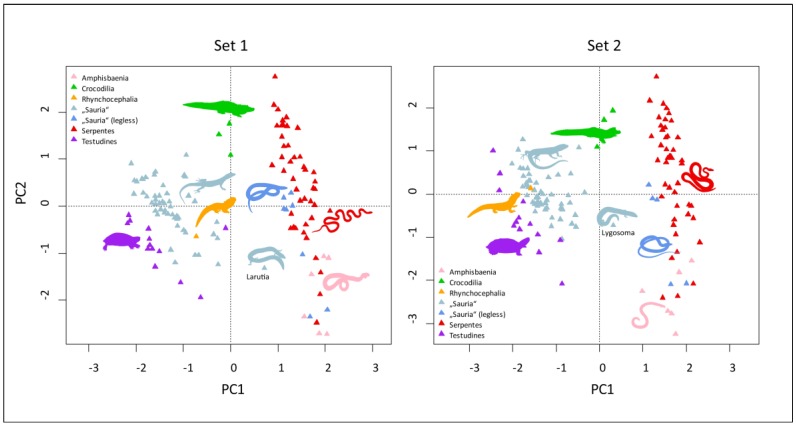
Plotting the species scores onto PC1 and PC2 axes, for Set 1 and Set 2 separately.

**Figure 3 animals-09-00238-f003:**
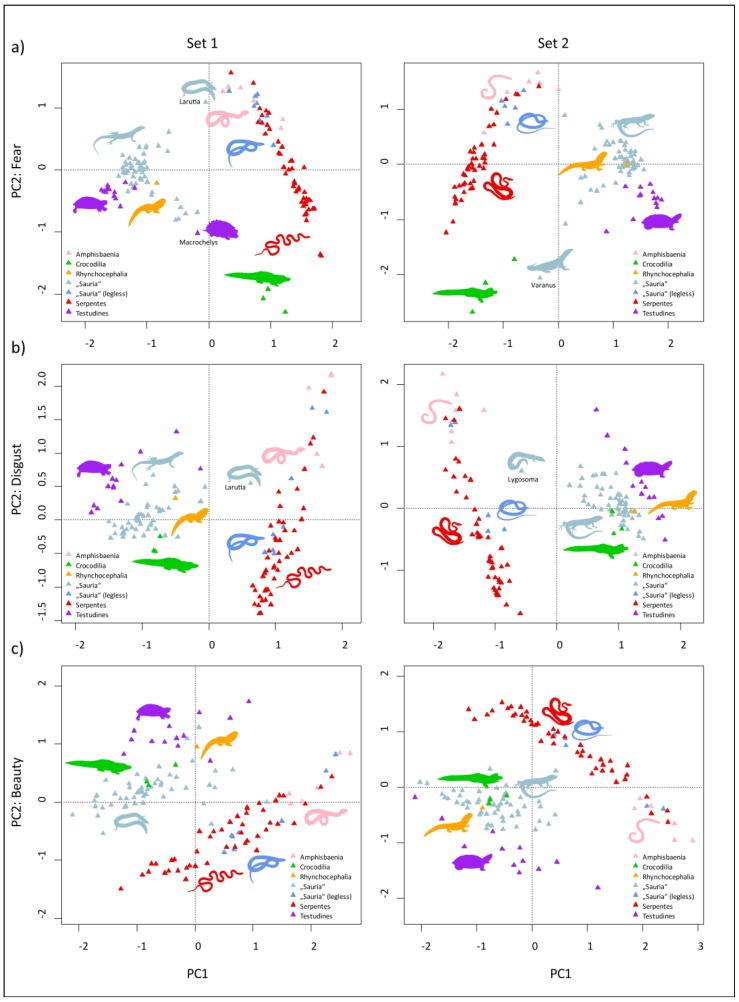
Plotting of the species scores onto PC1 and PC2 axes of the PC analyses, separately for (**a**) fear, (**b**) disgust, and (**c**) beauty rankings. In all cases, PC1 axis closely correlates with the respective mean values (e.g., PC1 fear correlates with mean fear), and the correlations are positive in Set 1 a, b, and negative in Set 1 c and Set 2 a, b, c. Please note that the positivity and negativity of the values of the PC axes are arbitrary and thus not fully comparable across the individual projections. For direct comparisons, please heed the colors that mark the specific taxa/groups and the distance of single points (triangles) and the groups they form from the other groups.

**Figure 4 animals-09-00238-f004:**
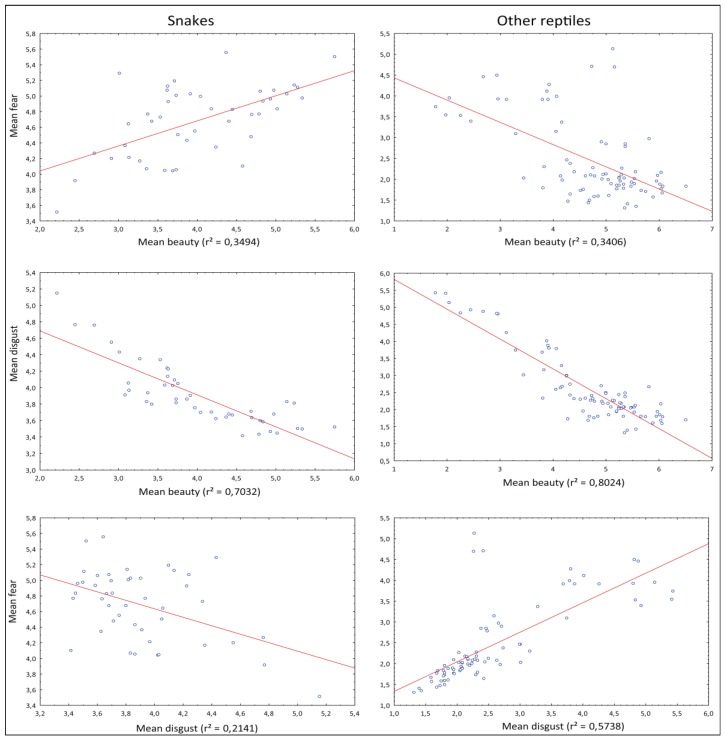
The correlations of the mean ranks for fear and beauty, disgust and beauty, fear and disgust; separately for snakes and other lizards.

**Table 1 animals-09-00238-t001:** Summary of the principal component (PC analysis of the datasets).

			Eigenvalue			Proportion Explained by		
Stimuli Subsets	No. of Axes	Eigenvalues > 1	PC1	PC2	PC3	PC1	PC2	PC3
Set1: fear	121	27	267.18	26.73	10.40	69.68 %	6.97 %	2.71 %
Set1: disgust	125	27	192.71	34.22	9.63	61.28 %	10.88 %	3.06 %
Set1: beauty	126	49	182.98	27.70	15.93	47.86 %	7.25 %	4.17 %
Set1: pooled	126	87	548.17	156.82	43.12	50.74 %	14.52 %	3.99 %
Set2: fear	126	38	388.81	33.04	13.01	70.94 %	6.03 %	2.37 %
Set2: disgust	126	33	229.10	41.94	10.32	61.19 %	11.20 %	2.76 %
Set2: beauty	126	56	188.36	44.20	19.55	42.53 %	9.98 %	4.41 %
Set2: pooled	126	103	696.29	193.68	52.10	51.00 %	14.18 %	3.82 %

**Table 2 animals-09-00238-t002:** Spearman correlations of the fear, disgust, and beauty rankings of the (**a**) Set 1 and (**b**) Set 2. Correlations significant at the *p* < 0.05 are marked in bold.

**(a) Set 1**	mean disgust	mean fear	mean beauty	PC1 fear	PC2 fear	PC1 disgust	PC2 disgust	PC1 beauty	PC2 beauty
mean disgust	**1**	**0.773**	**−0.836**	**0.786**	**0.369**	**0.995**	0.002	**0.859**	**−0.423**
mean fear	**0.773**	**1**	**−0.458**	**0.995**	−0.136	**0.796**	**−0.529**	**0.494**	**−0.682**
mean beauty	**−0.836**	**−0.458**	**1**	**−0.468**	**−0.471**	**−0.807**	**−0.333**	**−0.998**	0.030
PC1 fear	**0.786**	**0.995**	**−0.468**	**1**	−0.104	**0.808**	**−0.537**	**0.505**	**−0.711**
PC2 fear	**0.369**	−0.136	**−0.471**	−0.104	**1**	**0.354**	**0.408**	**0.475**	−0.031
PC1 disgust	**0.995**	**0.796**	**−0.807**	**0.808**	**0.354**	**1**	−0.043	**0.831**	**−0.454**
PC2 disgust	0.002	**−0.529**	**−0.333**	**−0.537**	**0.408**	−0.043	**1**	**0.296**	**0.719**
PC1 beauty	**0.859**	**0.494**	**−0.998**	**0.505**	**0.475**	**0.831**	**0.296**	**1**	−0.082
PC2 beauty	**−0.423**	**−0.682**	0.030	**−0.711**	−0.031	**−0.454**	**0.719**	−0.082	**1**
**(b) Set 2**	mean disgust	mean fear	mean beauty	PC1 fear	PC2 fear	PC1 disgust	PC2 disgust	PC1 beauty	PC2 beauty
mean disgust	**1**	**0.774**	**−0.797**	**−0.791**	**0.419**	**−0.988**	0.154	**0.853**	**0.423**
mean fear	**0.774**	**1**	**−0.421**	**−0.996**	−0.055	**−0.800**	**−0.346**	**0.504**	**0.724**
mean beauty	**−0.797**	**−0.421**	**1**	**0.428**	**−0.416**	**0.748**	**−0.451**	**−0.989**	0.041
PC1 fear	**−0.791**	**−0.996**	**0.428**	**1**	0.025	**0.817**	**0.353**	**−0.514**	**−0.744**
PC2 fear	**0.419**	−0.055	**−0.416**	0.025	**1**	**−0.421**	**0.402**	**0.425**	−0.064
PC1 disgust	**−0.988**	**−0.800**	**0.748**	**0.817**	**−0.421**	**1**	−0.095	**−0.811**	**−0.476**
PC2 disgust	0.154	**−0.346**	**−0.451**	**0.353**	**0.402**	−0.095	**1**	**0.374**	**−0.695**
PC1 beauty	**0.853**	**0.504**	**−0.989**	**−0.514**	**0.425**	**−0.811**	**0.374**	**1**	0.071
PC2 beauty	**0.423**	**0.724**	0.041	**−0.744**	−0.064	**−0.476**	**−0.695**	0.071	**1**

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
