# Peer review of "Human Attitude toward Reptiles: A Relationship between Fear, Disgust, and Aesthetic Preferences"

_animals, 2019, doi:10.3390/ani9050238_

Round 1

Reviewer 1 Report

 The  manuscript investigated the human perception of all reptiles focusing on the relationship between perceived fear, disgust, and aesthetic preferences and differences between snakes and other reptiles. It is a welcomed study of the issue.  It is well written and addresses an issue that has an obvious importance to an audience interested in herpetology, ethnozoology, ethnoherpetology and conservationists. I would suggest some  points to improve the manuscript. The manuscript is an ethnozoological/ethnoherpetological research. For this reason, I recommend that in introduction, authors explore more about relevance of ethnozoology and ethnoherpetology for animal/reptiles conservation.  I found interesting papers about these thematics on web using keywords: ethnozoology+conservation and ethnoherpetology + conservation (see references suggestions below).

            Results are well presented. In discussion, I suggest that authors improve the relevance of the study for reptiles conservation and sustainable use.

After these revisions, I think the manuscript should be accepted.

Some references suggestions

Alves, R. R. N. and W. M. S. Souto ( 2015). "Ethnozoology: A Brief Introduction." Ethnobiology And Conservation 4(1): 1-13.

Alves, R. R. N. (2012). "Relationships between fauna and people and the role of ethnozoology in animal conservation." Ethnobiology And Conservation 1: 1-69.

Mendonça, L. E. T., et al. (2014). "Caatinga Ethnoherpetology: Relationships between herpetofauna and people in a semiarid region." Amphibian & Reptile Conservation 8(1): 24–32.

Bertrand, H. (1997). "Contribution à l'étude de l'herpétologie et de l'ethnoherpétologie en Anjou= A study on the herpetology and ethnoherpetology of Anjou province (France)." Bulletin de la Société herpétologique de France(82-83): 51-62.

Das, I. (1998). The Serpent's tongue: a contribution to the ethnoherpetology of India and adjacent countries, Ed. Chimaira.

Fernandes-Ferreira, H., et al. (2013). "Hunting of herpetofauna in montane, coastal, and dryland areas of Northeastern Brazil." HERPETOLOGICAL CONSERVATION AND BIOLOGY 8(3): 652-666.

Alves, R. R. N., et al. (2012). "A zoological catalogue of hunted reptiles in the semiarid region of Brazil." Journal of Ethnobiology and Ethnomedicine 8(1): 27.

Speck, F. G. (1946). "Ethnoherpetology of the Catawba and Cherokee Indians." Wash. Acad. Sci., Jour 36: 355-360.

Author Response

The manuscript is an ethnozoological/ethnoherpetological research. For this reason, I recommend that in introduction, authors explore more about relevance of ethnozoology and ethnoherpetology for animal/reptiles conservation.

Response: Accepted. Thank you for this insightful comment, we have added a short section on ethnozoology and conservation into the Introduction: "The whole variety of human interactions with local fauna has been traditionally the subject of ethnozoology (or specifically ethnoherpetology when focused on reptiles), a discipline that has thrived since 19th century [15]. Recently, there is a growing body of evidence that ethnozoology through studying people’s perception of animals is indispensable in sustainable management of natural resources and might therefore play a crucial role in conservation of endangered species [16], including reptiles [17-19], protection of which is complicated by the aversion they often elicit in people [20]."

In discussion, I suggest that authors improve the relevance of the study for reptiles conservation and sustainable use.

Response: Accepted. Please see the final paragraph added to the Conclusions: "3) Our results might be used in conservation biology efforts invested in the management of reptile populations threatened by extinction. It has been previously shown that especially in the case of reptiles, the human perception and attitude plays a significant role in effectivity of local conservation programs. Reptiles, as opposed to other vertebrates such as mammals or birds, are not very popular and aversion they trigger might be a barrier to their protection. Here we show that particularly species with the snake-like body form are associated with intensive fear and disgust and therefore, this group might be susceptible to more human attacks and lesser conservation efforts. Previous research has shown that these negative emotions reflected in aversive attitudes might be overcome through various educational programs targeted on people living in the specific locality."

Reviewer 2 Report

This article reports interesting results about the perception of reptiles by humans. The method, sample size, and analyses are sound. The results are clear and the main conclusions are supported by the main findings. Notably, the fact that snakes can be perceived as beautiful and scary is particularly important. Some issues should be addressed however.

Major issues

On different instances, the authors claim that they embraced all reptiles. Although they actually included a wide variety of lineages, they did not include all species, and certainly not all morphotypes. Reducing speciose lineages (e.g. Elapids) with a single picture of a single species (Table S1 and S2) means that huge variation was omitted. Indeed, a king cobra is not similar to a coral snake (especially for human eyes), let alone marine snakes. Thus, I suggest tuning down this strong and repeated assertion that all reptiles were taken into account. Instead, it should be stated that at least one species representing the main lineages was used. Moreover, it is never indicated that the term reptile refers to a paraphyletic group (notably when extinct groups are considered).  Birds and snakes are more closely related to each other than to turtles. This major gap between popular knowledge and phylogeny should be clearly presented. In this study the authors mixed up both. In fact, the authors extracted phylogenetic information from the (very) paraphyletic reptile-group in order to please the popular vision through the use of the term reptiles (e.g. as if it was a well-identified clade that notably contains well-separated branches like chelonians, lizards, snakes or sphenodonts). I agree that the term reptile is convenient, and thus that it can be used, but the readers should be informed. As currently presented it is quite misleading.

In the same vein, the implicit taxonomic use of snakes versus lizards is not valid. Snakes truly belong to lizards. Anguis cephallonica is more closely related to snakes than it is to geckos for example. Thus, I suggest clarifying the fact that (semi-) popular knowledge (almost all reptile identification field guides frequently explain how to distinguish snakes from legless lizard, and thus make a taxonomic error) is taxonomically inaccurate. On the other hand, snakes are a peculiar group of lizard (e.g. long body and relatively short tail). Therefore, although it is valid to examine snakes as a group, it is invalid to oppose snakes and lizards. Consequently, it might be appropriate to define a group of lizards that visually appear to be snake-like: the legless reptiles (essentially snakes + other lizards).

The Liker scale used provides a mean to gauge to what extent species were ranked as frightening or not, for example, by the respondents. It would be useful to present the mean score of each group against meaningful references (i.e. mean value around <2 for species not considered as frightening; mean value around >5 for very scary species). The same information would be useful for disgust and beauty scores (on average, do people actually consider snakes, or crocodiles, as beautiful or not?).

Minor issues

Line 24: I do not understand the meaning of “and show what rules evaluation of such animals is conducted”

Line 31: “and a strong negative correlation between disgust and beauty and negative correlation between fear and beauty” could be simplified as “negative correlations among disgust, beauty and fear”.

Line 32: Consider deleting “However, the relationship between beauty, fear, and disgust evoked by snakes was different” as this sentence add no information.

Line 97: Please consider rephrasing “Some snakes can be either produce a potent..”

Lines 100-109: The authors should be prudent before stating that previous studies (e.g. refs 34-35) actually demonstrated humans were regularly preyed by snakes. All these studies, plus many others, are very speculative and should be handled cautiously. Thus, I suggest to start with something like “According to…”.

The two first questions at the end of the introduction are not clearly posed.

Line 142: “What is the relationship of people to reptiles as a group in terms of positive and negative emotions?” This sentence is obscure and should be reformulated.

Line 144: the meaning of “…as an evolutionary significant group” is also very obscure.

Please consider a single question instead of the two first. It is reasonable to examine if the gross morphology of the studied organisms influence how they are perceived by people; notably legged versus legless reptiles.

Line 150: how can we be sure that the selection of pictures was the one “that best represents the variability of all reptiles”? Instead I suspect that the pictures were selected opportunistically. Ontogenic changes, sexual dimorphism and geographic variations that are associated with considerable phenotypic variability were not implemented for example.

Line 163: did each person respond to the three series of questions? If so, why in the Tables S1 and S2 the respondent numbers are not corresponding (e.g. r1 can be woman with fear questions or a man with beauty questions? This is important to understand how the statistics were performed, especially the PCA.

Results and discussion: Regarding fear scores, the assertion that the snapping turtle and Komodo dragon were clearly separated from the non-snake reptiles and also closer to snakes is not obvious. The snapping turtle is indeed detached from other chelonians but it remains within its “fear group”. Crocodile are more clearly different, especially regarding fear, and the Komodo dragon is more crocodile like than snake like in the projection of the PCI axes. This should be clarified.

Figure 3: the respective positions of snakes and other reptiles are changing, presumably due to different computations during PCA (e.g. 2 top panels, snakes are either with positive or negative values on the PC1). Because the projections are partly arbitrary, it would be convenient to make more uniform representations in order to facilitate comparisons.  

Line 301: “…although fatal attacks from these species are rare [83]”. In fact the ref cited shows that the attack was rather benign. This quotation is thus misleading. More generally, in contrast to what is stated, humans are not preyed by turtles or monitor lizards. Predation by snakes is exceptional while humans eat millions of snakes per year in tropical countries. Large crocodiles (not all species) are the sole reptiles that consider humans as prey. This should be clarified. Venomous snakes represent a real health burden, but not at all turtles (even giant) and monitors (even large).

309: Comodo or Komodo?

Line 342: I don’t understand one conclusion: “Simultaneously, they have a specific bodily plan which guides the human’s categorization process”; indeed, simultaneously with what? Do you refer to the previous sentence? This is confusing. Consider deleting “simultaneously”.

Author Response

Major issues

On different instances, the authors claim that they embraced all reptiles. Although they actually included a wide variety of lineages, they did not include all species, and certainly not all morphotypes. Reducing speciose lineages (e.g. Elapids) with a single picture of a single species (Table S1 and S2) means that huge variation was omitted. Indeed, a king cobra is not similar to a coral snake (especially for human eyes), let alone marine snakes. Thus, I suggest tuning down this strong and repeated assertion that all reptiles were taken into account. Instead, it should be stated that at least one species representing the main lineages was used. Moreover, it is never indicated that the term reptile refers to a paraphyletic group (notably when extinct groups are considered).  Birds and snakes are more closely related to each other than to turtles. This major gap between popular knowledge and phylogeny should be clearly presented. In this study the authors mixed up both. In fact, the authors extracted phylogenetic information from the (very) paraphyletic reptile-group in order to please the popular vision through the use of the term reptiles (e.g. as if it was a well-identified clade that notably contains well-separated branches like chelonians, lizards, snakes or sphenodonts). I agree that the term reptile is convenient, and thus that it can be used, but the readers should be informed. As currently presented it is quite misleading.

Response: ACCEPTED. The inclusion of “all reptiles” was in relation to other studies (which usually include only lizards) and other animal taxa (mammals and birds), which are usually represented by many more species. We agree that this was misleading and we corrected the Methods section as well as Introduction as suggested by the reviewer:

“To our knowledge, no one has ever studied the human perception of reptiles in whole, as a category that would include representatives of different taxonomic clades (i.e., turtles and tortoises, lizards, crocodiles, and tuatara) reflecting their large-scale biological diversity.”

“To test the relationship of people to reptiles, we have prepared two sets of pictures that cover a wide variability of the reptile morphology: at least one species representing the main lineages was used. These sets were designed to cover the wide variability while retaining a reasonable amount of stimuli that could be rated by respondents without substantial tiredness. We…”

We also rewrote the Introduction (made changes throughout the whole text) so now it takes into account that reptiles form a paraphyletic group and that snakes are part of the lizards. Most importantly, we added the following text to the beginning of the Introduction:

“Reptilia represents an ancient and diversified group of vertebrates that includes Testudines (turtles, tortoises and terrapins) and Diapsida, which further splits into Lepidosauria (comprising Squamata and Rhynchocephalia) and Archosauria (comprising Crocodylia and Aves). The precise position of Testudines within Diapsida is uncertain (Rieppel, 1996). They may be a sister groupof either lepidosaurs (Lyson et al., 2011) or archosaurs (Chiari et al.2012). The birds, however, differ so much from the other groups that they are almost always recognized and categorized as a separate category by humans, both in traditional (Linnæus, 1758) and ethnobiological classification (Berlin, 2014). Because this paper is focused on human perception of these animals, we will use the term “reptiles” in the same manner, i.e., as a paraphyletic group of Reptilia excluding birds and extinct species.”

In the same vein, the implicit taxonomic use of snakes versus lizards is not valid. Snakes truly belong to lizards. Anguis cephallonica is more closely related to snakes than it is to geckos for example. Thus, I suggest clarifying the fact that (semi-) popular knowledge (almost all reptile identification field guides frequently explain how to distinguish snakes from legless lizard, and thus make a taxonomic error) is taxonomically inaccurate. On the other hand, snakes are a peculiar group of lizard (e.g. long body and relatively short tail). Therefore, although it is valid to examine snakes as a group, it is invalid to oppose snakes and lizards. Consequently, it might be appropriate to define a group of lizards that visually appear to be snake-like: the legless reptiles (essentially snakes + other lizards).

Response: ACCEPTED. The inclusion of “all reptiles” was in relation to other studies (which usually include only lizards) and other animal taxa (mammals and birds), which are usually represented by many more species. We agree that this was misleading and we corrected the Methods section as well as Introduction as suggested by the reviewer:

We corrected this within the text and we also added a more detailed description of why snakes may represent an important cognitive category that is worth examining outside of the lizards group:

“Conversely, a group of reptiles is most often represented by only a single taxonomic category, i.e., the lizards (including snakes), most often further unspecified [11] or represented by the common sand lizard (Lacerta agilis) and grass snake (Natrix natrix) or some pythons (Python sp.)”

“This phenomenon is supported by studies showing a rapid detection of snakes compared to inanimate objects such as flowers and mushrooms [39-42] or other animals [43-45], mediated by specific neural mechanisms (Isbell, 2006; LoBue and DeLoache, 2008; Öhman et al., 2012; Van Le et al., 2013; Baynes-Rock, 2017). This was reported in both humans and non-human primates (Shibasaki & Kawai, 2009; Kawai & Koda, 2016). Because of these reasons, it is rational to believe that snakes, although taxonomically a part of lizards, represent a different cognitive category for humans. As a substantial part of this study we aimed to examine this specific position of snakes in relation to fear, but also to disgust and beauty (as a positive dimension to contrast with fear).”

The Liker scale used provides a mean to gauge to what extent species were ranked as frightening or not, for example, by the respondents. It would be useful to present the mean score of each group against meaningful references (i.e. mean value around <2 for species not considered as frightening; mean value around >5 for very scary species). The same information would be useful for disgust and beauty scores (on average, do people actually consider snakes, or crocodiles, as beautiful or not?).

Response: ACCEPTED. Table 1 and 2 in Supplements now includes mean ratings for fear, disgust, and beauty for each group together with a lower and higher quartile calculated from the whole set for each dimension as a reference value.

Minor issues

Line 24: I do not understand the meaning of “and show what rules evaluation of such animals is conducted”

Response: ACCEPTED. The sentence has been rephrased: "...and show the key feautures that our evaluation of such animals is based on."

Line 31: “and a strong negative correlation between disgust and beauty and negative correlation between fear and beauty” could be simplified as “negative correlations among disgust, beauty and fear”.

Response: ACCEPTED. The sentence has been rephrased: "... while disgust and fear were both negatively correlated with beauty."

Line 32: Consider deleting “However, the relationship between beauty, fear, and disgust evoked by snakes was different” as this sentence add no information.

Response: ACCEPTED. The sentence has been removed.

Line 97: Please consider rephrasing “Some snakes can be either produce a potent..” 

Response: ACCEPTED. The whole sentence has been rephrased: "Certain snakes, on the other hand, can either produce potent venom capable of killing an adult human (Viperidae, Elapidae; [31]) or become dangerous when reaching critical size, especially large constrictors like the reticulated python (Malayopython reticulatus, [32]) or green anaconda (Eunectes murinus, [33] (p.131-164)).

Lines 100-109: The authors should be prudent before stating that previous studies (e.g. refs 34-35) actually demonstrated humans were regularly preyed by snakes. All these studies, plus many others, are very speculative and should be handled cautiously. Thus, I suggest to start with something like “According to…”.

Response: ACCEPTED. The sentence has been changed as suggested: "According to some authors [e.g., 34,35],..."

The two first questions at the end of the introduction are not clearly posed.

Line 142: “What is the relationship of people to reptiles as a group in terms of positive and negative emotions?” This sentence is obscure and should be reformulated.

Line 144: the meaning of “…as an evolutionary significant group” is also very obscure.

Please consider a single question instead of the two first. It is reasonable to examine if the gross morphology of the studied organisms influence how they are perceived by people; notably legged versus legless reptiles.

Response: ACCEPTED. The first two questions have been merged and rephrased: "What is the human attitude toward a group of reptiles in terms of positive and negative emotions and does our evaluation rely on basic morphotypes, specifically, are legless snakes perceived differently than the rest of reptiles?"

Line 150: how can we be sure that the selection of pictures was the one “that best represents the variability of all reptiles”? Instead I suspect that the pictures were selected opportunistically. Ontogenic changes, sexual dimorphism and geographic variations that are associated with considerable phenotypic variability were not implemented for example.

Response: ACCEPTED. We changed this part of the Methods to better describe our aim, i.e., to cover a wide variability of reptiles (especially when compared to other papers that only included a few species belonging to lizards or turtles).

“To test the relationship of people to reptiles, we have prepared two sets of pictures that cover a wide variability of all reptile morphology: at least one species representing the main lineages was used. These sets were designed to cover the wide variability while retaining a reasonable amount of stimuli that could be rated by respondents without substantial tiredness. We…”

Line 163: did each person respond to the three series of questions? If so, why in the Tables S1 and S2 the respondent numbers are not corresponding (e.g. r1 can be woman with fear questions or a man with beauty questions? This is important to understand how the statistics were performed, especially the PCA.

Response: ACCEPTED/EXPLAINED. As already stated in the text, regarding the three dimensions: “Some respondents evaluated the sets by all of the measured dimensions, however, because there was a reasonable time delay between each of the evaluations (several months), we considered them independent.”

However, it was not clearly stated whether both of the Sets (Set 1 and Set 2) were evaluated by the same respondents. Thus, we added this detail to the Materials and methods section to clarify this:

“Each of the sets (Set 1 and 2) were evaluated by different respondents: Set 1 evaluated 122 respondents by perceived fear, 126 by disgust, and 157 by beauty; set 2 evaluated 184 (fear), 143 (disgust), and 188 (beauty) respondents.”

Results and discussion: Regarding fear scores, the assertion that the snapping turtle and Komodo dragon were clearly separated from the non-snake reptiles and also closer to snakes is not obvious. The snapping turtle is indeed detached from other chelonians but it remains within its “fear group”. Crocodile are more clearly different, especially regarding fear, and the Komodo dragon is more crocodile like than snake like in the projection of the PCI axes. This should be clarified.

Response: ACCEPTED. Thank you for pointing this out. We rewrote the Results and Discussion to clarify this. Also, the possible reasons for this observation were removed from Results and extended within the Discussion.

The text was rewritten as follows:

“(…) The most pronounced difference was that in the case of fear, crocodiles formed a distinct group. Moreover, the Komodo dragon (Varanus komodoensis, Set 2) split from the main group of “other reptiles” and joined closer to the crocodiles. The alligator snapping turtle (Macrochelys temminckii, Set 1) split off from the other Testudines and joined closer to the crocodiles, but still remained within the main group of “other reptiles. (…)”

 “When reptiles are evaluated on specific dimensions (fear, disgust, and beauty), the two previously mentioned groups are almost always formed. The only exception is the evaluation of fear, in which crocodiles form a separate group. (…) There is also the alligator snapping turtle (Macrochelys temminckii) and Komodo dragon (Varanus komodoensis), which may be potentially dangerous for humans considering their large body size, although attacks from these species are very rare and the inflicted injuries are mostly manageable [83]. In fact, contrary to crocodiles or large constrictors, both the snapping turtle and Komodo dragon do not consider humans as prey. It is thus possible that these animals evoke high fear in the respondents not due to their real dangerousness but rather due to their physical appearance, which may include large body size, sharp edges, and dark color. ”

Figure 3: the respective positions of snakes and other reptiles are changing, presumably due to different computations during PCA (e.g. 2 top panels, snakes are either with positive or negative values on the PC1). Because the projections are partly arbitrary, it would be convenient to make more uniform representations in order to facilitate comparisons. 

Response: ACCEPTED/EXPLAINED. Indeed, the respective positions of snakes and other reptiles are changing because the projections are arbitrary. Because of that, we marked all of the separate taxonomic groups specific colors that are uniform across the different projections, mainly to make easier and more convenient comparisons possible. You are right, however, that this was poorly described in the figure caption and thus we edited the figure caption to make this clearer:

“Figure 3. Plotting of the species scores onto PC1 and PC2 axes of the PC analyses, separately for a) fear, b) disgust, and c) beauty rankings. In all cases, PC1 axis closely correlates with the respective mean values (e.g., PC1 fear correlates with mean fear), and the correlations are positive in Set 1 a, b, and negative in Set 1 c and Set 2 a, b, c. Please note that the positivity and negativity of the values of the PC axes are arbitrary and thus not fully comparable across the individual projections. For direct comparisons, please heed the colors that mark the specific taxa/groups and the distance of single points (triangles) and the groups they form from the other groups.”

We believe that this will make the comparison more convenient. Unfortunately, we do not know any better projection that would allow for both statistical correctness and unified projection of the results. Another option could be a schematic figure, but we believe that the direct projection of the results of the PC analyses is better in this case.

Line 301: “…although fatal attacks from these species are rare [83]”. In fact the ref cited shows that the attack was rather benign. This quotation is thus misleading. More generally, in contrast to what is stated, humans are not preyed by turtles or monitor lizards. Predation by snakes is exceptional while humans eat millions of snakes per year in tropical countries. Large crocodiles (not all species) are the sole reptiles that consider humans as prey. This should be clarified. Venomous snakes represent a real health burden, but not at all turtles (even giant) and monitors (even large).

Response: ACCEPTED. The sentence has been rephrased and another one added: "There is also the alligator snapping turtle (Macrochelys temminckii) and Komodo dragon (Varanus komodoensis), which may be potentially dangerous for humans considering their large body size, although attacks from these species are very rare and inflicted injuries are mostly manageable [83]. In fact, contrary to crocodiles or large constrictors, both the snapping turtle and Komodo dragon do not consider humans as prey."

309: Comodo or Komodo?

Response: ACCEPTED. We corrected it to Komodo.

Line 342: I don’t understand one conclusion: “Simultaneously, they have a specific bodily plan which guides the human’s categorization process”; indeed, simultaneously with what? Do you refer to the previous sentence? This is confusing. Consider deleting “simultaneously”.

Response: ACCEPTED. The sentence has been rephrased and merged with the previous one: "1) Snakes are such important and specific animals that they are completely separated from the other reptiles as regards our perception of them, especially fear. We hypothesize that it is their peculiar legless body plan that guides the human categorization process. Moreover, the same pattern expands onto other species resembling snakes, the worm lizards (Amphisbaenia) and legless lizards."

Reviewer 3 Report

I think this is an innovative and interesting paper. However, the methods need to be clarified and the pictures made available. On the linked website, it seems the only way to see the pictures is to take the test. Work by

In the results I would like to actually have available the mean preference scores for each species on each dimension. We also know that the posture of animals can play a large role. Take the work of Nobuo Masataka on snakes. The snapping turtle with open mouth could clearly bias responses. Thus, I am unsure that some of the species statements are correct. Perhaps very comparable photos are needed throughout, but I could not evaluate them. A size scale might also be useful as size may affect fear, though not disgust or beauty.

How the scoring was carried out needs more detail also. We need tables, at least on an accessible supplementary link, of the scores for all dimensions for all species. It is unclear whether the provided tables have this information as they were not annotated properly. It is also important the the statistics are transparent and vetted by an expert.

Note that it is Komodo dragon.

The findings, however, are very interesting if they remain valid after the above are addressed. These may be more specific discussion of findings by other researchers who have included reptiles in ratings of species. Though not part of this study, I wonder if any meta-analysis has been carried out on this work.

Author Response

In the results I would like to actually have available the mean preference scores for each species on each dimension. We also know that the posture of animals can play a large role. Take the work of Nobuo Masataka on snakes. The snapping turtle with open mouth could clearly bias responses. Thus, I am unsure that some of the species statements are correct. Perhaps very comparable photos are needed throughout, but I could not evaluate them. A size scale might also be useful as size may affect fear, though not disgust or beauty.

ACCEPTED. We added new sheets (marked a) into the SM1 and SM2, in which you can find mean preference scores for each species on each dimension (and a list of all used sources).

As for the photos, we could not include them within the manuscript due to copyright restrictions. However, we added the full list of all included sources to the Supplementary Materials 1 and 2 (part f).

Also, we extended the Discussion with emphasis on possible effect of the posture:

“The shape of an animal allometrically changes with changes in the body size. Because of this, human respondents are able to estimate size [27-31,79-81] and age (on the level of young/adult; [103]) of animals even when presented on pictures standardized for size. Similar effect was observed in this study. Larger reptile species were evaluated as more frightening, including the Komodo dragon. Moreover, as shown on snake studies, posture of the animal can affect both rapid detection [104] and evaluation of fear [80]. In this study, the alligator snapping turtle was pictured with an open mouth, showing its sharp beak, which might have elicited a feeling of a threat within the respondents. Another explanation might be that the turtle was rated as frightening because of a general perceptual bias towards sharp objects; it has been reported in literature that people rate sharp objects as more dangerous [105] and also activate greater neural response in amygdala when watching sharp objects [106].”

How the scoring was carried out needs more detail also. We need tables, at least on an accessible supplementary link, of the scores for all dimensions for all species. It is unclear whether the provided tables have this information as they were not annotated properly. It is also important the the statistics are transparent and vetted by an expert.

ACCEPTED/EXPLAINED. Tables with all the scores for all dimensions were already uploaded with the submission, however, they were not described properly. Thus, we added a detailed description of the included data. Moreover, since more data were added (mean preference scores for each species for each dimension and a list of all used sources), we divided them into separate sheets marked by letters a-f.

Note that it is Komodo dragon.

ACCEPTED.

Round 2

Reviewer 1 Report

Thank you for your attention to the comments from my previous review. Now, in my opinion, the manuscript is ready for publication.

Reviewer 3 Report

Basically fine now as authors responded well to the reviewer comments.